# Effect of Mo on the Microstructures and Mechanical Properties of the Polycrystalline Superalloy with High W Content

**DOI:** 10.3390/ma15217509

**Published:** 2022-10-26

**Authors:** Qiongrui Quan, Shijie Sun, Naicheng Sheng, Juan Deng, Guichen Hou, Jinguo Li, Jidong Chen, Yizhou Zhou, Xiaofeng Sun

**Affiliations:** 1AECC Aero Science and Technology Co., Ltd., Chengdu 61000, China; 2Shi-changxu Innovation Center for Advanced Materials, Institute of Metal Research, Chinese Academy of Sciences, Shenyang 110016, China

**Keywords:** high-W superalloy, Mo content, TCP phase, mechanical properties

## Abstract

The effect of the Mo contents of 1.0 wt.%, 1.5 wt.%, 2.0 wt.%, and 3.0 wt.% on the microstructures and mechanical properties of the polycrystalline superalloy with a high W content was studied. The typical dendrite morphology was observed in the high-W superalloy with different Mo contents, containing γ matrix, γ′ phase, eutectic, and MC carbide. After the heat treatment, the primary MC carbides were decomposed into M_6_C carbides, while a needle-like topologically close-packed (TCP) phase was formed in the alloy with high Mo content, in contrast to the other three alloys with low Mo content. The Mo addition increased the lattice parameter of the γ and γ′ phases and also changed the lattice misfits of the γ and γ′ phase lattice misfits towards a larger negative. The addition of Mo improved the yield strength at room temperature due to the solid solution strengthening and coherency strengthening. The improvement of the stress rupture lives at 975 °C/225 MPa was due to the combination of the suppressed propagation of the microcracks by the carbides and a more negative misfit. When the Mo content reached 3.0 wt.%, the TCP phases formed and decreased the ultimate tensile strength and the stress rupture lives as a result.

## 1. Introduction

Ni-based cast superalloys have become the preferred materials for turbine blades of aerospace engines because of their excellent comprehensive properties at high temperature [1,2,3]. In recent years, the service temperature of turbine blades has been getting higher due to the increasing thrust-to-weight ratio of aero-engines. Hence, the contents of solid solution strengthening elements, i.e., W and Mo, and precipitation strengthening elements, i.e., Al and Ti, continue to increase [4,5,6,7]. Recently, high W-content, Ni-based superalloys have been widely used in the production of engine turbine blades because of their low cost and high temperature-bearing capacity [8,9].

High-W superalloys, such as Mar-M 247, ЖC6У, and K416B, possess complex composition traits, containing a large amount of refractory element W, which acts as a solid solution strengthener and increases the fraction of fine γ′ precipitates [10,11]. Furthermore, the diffusion ability of W solute is very low due to the large atomic radius and atomic mass, which could suppress the diffusion of the other strengthening elements in the alloy, improving the structural stability and the creep resistance [12,13]. Meanwhile, the inexpensive metal W could replace Re and Ru to a certain extent, which would reduce the cost of the superalloys. However, macrosegregation is prone to appear in the high-W superalloys due to the large solidification segregation coefficient of the W element, which would lead to the phenomenon of uneven composition in the alloy, especially in the large-size ingots during the industrial production process, while it is difficult to eliminate the macrosegregation of the alloy through the heat treatment process because of the low diffusion coefficient of W [14,15]. In addition, it is easy to precipitate the W-rich phase, such as α-W and M_6_C during the solidification process, which would consume a large amount of W in the alloy, resulting in the reduction in the solid solution strengthening ability of W [16].

As compared with W, Mo has a similar strengthening effect, while Mo tends to distribute in the γ phase and promote the denser γ/γ′ interfacial dislocation networks during the high temperature creep test, which is beneficial to the improvement of the creep properties [17,18]. Furthermore, Mo has lower density relative to W, and substituting Mo for W would contribute to the decrease in the density of the superalloys, which could be conducive to the improvement of the thrust-to-weight ratio of advanced aero-engines. Hence, some new types of Mo-rich superalloys were developed, and it was also found that the microsegregation of the alloying elements and the tendency of the forming freckle during solidification could be decreased by substituting Mo for W [19,20], while the topologically close-packed (TCP) phases were usually precipitated in the single-crystal superalloys with high Mo content during the creep test, which would degrade the creep properties of the superalloys. Some researchers explored the optimal content of the Mo element for the different types of single-crystal superalloys [21,22,23,24]. However, there is little further detailed and systemic research on the effect of Mo on the polycrystalline superalloy with high W content. The target of this research is to provide more information for developing high-strength, high-W superalloys.

## 2. Materials and Methods

### 2.1. Materials Preparation

In this paper, the master ingot of high-W superalloy without the Mo element was prepared by vacuum induction melting (VIM), and then, during the casting process, Mo was added into the alloy with four different concentrations: 1.0 wt.%, 1.5 wt.%, 2.0 wt.%, and 3.0 wt.%. The chemical composition is listed in Table 1, and the alloys with different Mo contents are labeled as Alloy I, II, III, and IV; the other elements, except Mo, were equivalent in the four high-W superalloys. Then, the equiaxed crystal bars were obtained. Subsequently, the cast bars were subjected to the solution heat treatment at 1210 °C for 4 h, followed by air cooling. Meanwhile, in order to determine the lattice misfit of the high-W superalloy with different Mo contents, the single-crystal bars were also produced by the Bridgman high-rate solidification technique under a high thermal gradient in a vacuum furnace using the same chemical compositions.

### 2.2. Microstructural Observation

A cylindrical specimen was cut from the cast and heat-treated bars and then mechanically ground, polished, and chemically etched in a solution of 20 g CuSO_4_ + 100 mL HCl + 5 mL H_2_SO_4_ + 80 mL H_2_O for microstructural observation. The microstructures of the high-W superalloys with different Mo contents were observed by Gemini SEM 300 field emission scanning electron microscope (FE-SEM) with the Ultimax 65 detector. Specimens with a diameter of 5 mm and a height of 3 mm were prepared for SEM observations. Thin slices were cut for TCP phase observation by a transmission electron microscope (TEM). The slices were first mechanically ground to 50 µm thick, then twin-jet thinned in a solution of 10% perchloric acid and 90% alcohol at −20 °C by Tenupole-5. TEM observation was performed on TECNAI F20. The intensity profiles of the (004) reflection of the single-crystal superalloys were measured by the Bruker D8 Discover. The lattice parameters and misfit of the γ and γ′ phases were calculated.

### 2.3. Mechanical Properties

In order to evaluate the effect of the Mo content on the mechanical properties, the tensile tests performed at room temperature and the stress rupture tests performed at 975 °C/225 MPa in air were conducted. Specimens with a diameter of 5 mm and a gage length of 30 mm were machined for the tensile tests, and specimens with a diameter of 5 mm and a gauge length of 25 mm were machined for the stress rupture tests. Three specimens were tested for each alloy for statistical significance. After mechanical tests, the fracture surfaces were examined by SEM to analyze the fracture mode. Longitudinal section samples with a length of 10 mm, a width of 4 mm, and a thickness of 1 mm were cut from the fractured specimens. After grinding, polishing, and chemical etching, the longitudinal sections were observed by SEM to study the microstructure evolution and deformation mechanism of the superalloy.

## 3. Results and Discussion

### 3.1. Microstructures

The cast microstructures of the high-W superalloys with different Mo contents are shown in Figure 1. The typical dendrite morphology was observed in the alloy which contained γ matrix, γ′ phase, eutectic, and MC carbides. With the increase in the Mo content, the amount of carbides in the alloy increased slightly, which was because that Mo element is the MC carbide-forming element, as shown in Figure 1f. The γ′ phase was fine in the dendrite core, while the γ′ phase was coarse in the interdendritic region. There was little change of the size of γ′ phase with the increase in Mo content, and it was ~1.3 μm in the dendrite core. Generally, the content of eutectic in the alloy is related to the degree of segregation of the elements during the solidification process [25]. Due to the increased Mo element that formed the carbides or dissolved into the matrix, the forming elements of the eutectic Al or Ti in the interdendritic region were not affected. Thus, the number of eutectics in the alloy did not change significantly with the increase in Mo content.

Figure 2 shows the microstructures of alloys with different Mo content after heat treatment. The microstructures contained fine γ′ phases, the gray-contrast MC carbides, and the white-contrast M_6_C carbides in the heat-treated alloys. Generally, the MC carbide can be decomposed into M_6_C carbides during the heat-treatment process because MC carbide is less stable than M_6_C carbide at high temperature [26]. In addition, the primary MC carbides were enriched in the high levels of Ti, Mo, Nb, and W, while the M_6_C carbides were enriched in the high levels of W and Mo, as shown in Figure 2e,g,h. As can be seen from Figure 2d, a new phase was formed in Alloy IV in contrast to the other three alloys with low Mo content after the heat treatment, which was a needle-like structure in morphology and enriched in W and Mo elements. The selected area diffraction pattern (SADP) (Figure 2d) identified that it was a μ phase. It was also found that the TCP phases usually formed at the interdendritic region, as indicated by the arrows in Figure 2d.

The elemental segregation behavior between the dendritic and interdendritic regions in the high-W superalloys with various Mo was characterized. The values of the elemental segregation coefficient (*k*) were defined as the ratio of the average concentration of the alloying elements in the dendrite core to that in the interdendritic region:(1)$$k=\frac{x_{i,Dendrite}}{x_{i,\;Interdendritic}}$$
where *x_i,Dendrite_* and *x_i,Interdendritic_* are the concentrations of each specific element for the dendrite core and the interdendritic region. A *k* value of less than 1 indicates segregation to the interdendritic region, while a *k* value of greater than 1 indicates segregation to the dendrite core [27]. It is seen that the segregation behaviors of the different elements in the alloys are obviously different, as shown in Figure 3. There was a slight segregation of the Cr, Co, and Ni elements. In contrast, Al, Ti, Nb, and Mo tended to concentrate in the interdendritic region, and W tended to segregate in the dendrite core. It was also found that the degree of segregation of the W element that typically partitioned to the dendrite core decreased when molybdenum was added to the alloy. Similar results were also reported in the previous work [27]. The segregation coefficient of the elements was decreased after the heat treatment, as shown in Figure 3b.

Generally, the TCP precipitates are easily formed in the microstructures of Ni-based superalloys during the high-temperature, long-term aging or creep tests, which is essentially caused by the element diffusion and has a great relationship with temperature, time, and stress [28,29]. W, Mo, and the other refractory elements are usually enriched in the γ matrix, and the segregation of the above elements on the dendrite core cannot be completely eliminated during the solid solution treatment, thus the TCP phases precipitate in the dendrite core [30,31]. In contrast, the TCP phases formed at the interdendritic region in Alloy IV with 3.0 wt.% Mo content after the heat treatment in this work, which could be explained by the following reasons:

Mo is the strong forming element for the TCP phase; thus, the formation tendency of the TCP phases increased with the increase in Mo content. While no TCP phase was observed in the cast microstructure of Alloy IV, perhaps due to the Mo segregated in the interdendritic region, the W segregated in the dendrite core, as shown in Figure 4a, and the refractory elements saturation limit for the formation of TCP was not exceeded. The W element diffused to the interdendritic region during the heat-treatment process, and the refractory concentration exceeded the saturation limit for the formation of TCP, which induced the formation of TCP phases in the interdendritic region, as shown in Figure 4b.

The lattice parameter of the γ and γ′ phases of the high-W superalloys with different Mo contents were determined by XRD. The (004) two-phase reflection was chosen for deconvolution and determination of the lattice parameters of the two phases because of the better signal-to-noise ratio and higher detected data accuracy provided by the high-order peaks [32]. Figure 5 shows the (004) diffraction reflections of the high-W superalloys with different Mo contents. The sharp diffraction peak exhibited in Figure 5a indicates a strong overlap between the individual phases of γ and γ′, demonstrating a low misfit between the γ and γ′ phases in Alloy I [33]. With the increase in the Mo content, the width of the diffraction peaks increased gradually, which indicated that the misfit between the γ and γ′ phases increased gradually. Furthermore, the diffraction peaks shifted to the left, which demonstrated that the lattice parameter of the γ and γ′ phases increased gradually, as shown in Figure 5b–d. The lattice parameter of the γ and γ′ phases for the high-W superalloys were calculated by the Bragg equation, and the results are shown in Figure 6a. It was found that the lattice parameter of the γ and γ′ phases increased gradually with the increase in the Mo content. In contrast to the γ′ phase, the lattice parameter of the γ phase increased quickly, which indicated that the misfit between the γ and γ′ phases increased gradually. The misfit (*δ*) between the γ and γ′ phases could be calculated by the following equation [34]:(2)$$\delta =\frac{2\left(a_{\gamma^{\prime }}-a_{\gamma }\right)}{a_{\gamma^{\prime }}+a_{\gamma }}$$
where *a*_γ′_ and *a*_γ_ are the lattice parameters of the γ′ and γ phases, and the calculated results are shown in Figure 6b. It could be seen that the misfit between the γ and γ′ phases was negative, which is the same as for the other Ni-based superalloy [35].

### 3.2. Mechanical Properties

Figure 7 shows the yield strength, ultimate tensile strength, and elongation of the high-W superalloys with various Mo contents tested at room temperature. It was found that with the increase in the Mo content, the yield strength of the alloys increased and the elongation decreased, while the ultimate tensile strength increased first and then decreased, with the maximum value occurring at 2 wt.% Mo. The fracture surfaces of the high-W superalloys are shown in Figure 8. It was clearly seen that the dendrite morphology could be observed in the fracture surfaces, which suggested that the cracks mainly propagated along the interdendritic region during the tensile test, as shown in Figure 8a–d. As can be seen from Figure 8e–h, the cracks initiated from the eutectic and propagated along the interdendritic region.

The increase in yield strength of the high-W superalloys with the increase in the Mo content at room temperature was mainly attributed to the solid solution strengthening and coherency strengthening. The solid solution strength of the γ matrix contributed by each element can be given as [36,37]:(3)$$S_{i}^{\gamma }=\beta_{i}^{\gamma }{\left(x_{i}^{\gamma }\right)}^{1/2}$$
where $\beta_{i}^{\gamma }$ is a constant related to the atom radius and modulus of the element *i*, and $x_{i}^{\gamma }$ is the content of element *i* in the γ matrix. Thereby, the contribution of the solid solution strength to the yield strength would be enhanced with the increase in the Mo content dissolved in the matrix. The strengthening effect arising from the lattice misfit (*τ*_coh_) was used to determine the coherency strengthening, using Equation (4) [38]:(4)$$\tau_{\mathrm{coh}}=\alpha \mu {\left|\delta \right|}^{3/2}{\left[\frac{fr}{b}\right]}^{1/2}$$
where *δ* is the lattice misfit, *r* is the average precipitate radius, and *α* is a constant. As the absolute value of *δ* was increased with the addition of the Mo element, the contribution of the coherency strengthening was enhanced. The strengthening mechanism in the ultimate tensile strength was similar to the yield strength. Furthermore, more Mo elements were segregated in the interdendritic region with the increase in the Mo content; thus, the ultimate tensile strength of the alloy was improved to a certain extent as the strength of the interdendritic region improved. However, a large number of TCP phases precipitated in the microstructures with the Mo content increase to 3.0 wt.%, consuming the solid solution strengthening elements W and Mo, which resulted in the decrease in the strength of the interdendritic region. The micro-cracks were more likely to form and propagate at the interdendritic region during the tensile test, which resulted in the premature failure in Alloy IV.

The stress rupture tests of the high-W superalloys with various Mo contents were carried out at 975 °C/225 MPa, and the results are shown in Figure 9. The stress rupture lives exhibited a very steep peak as a function of the Mo content, with the maximum in stress rupture life of 60 h occurring at about 2.0 wt.% Mo. However, as the Mo content continued to increase to 3.0 wt.%, the stress rupture life sharply decreased to 36 h. It could be inferred that the addition of Mo improved the stress rupture properties of the high-W superalloy but that it will reduce the stress rupture properties when excessive Mo is added. The reasons for the differences in the stress rupture properties of the high-W superalloys with various Mo contents are explained below.

Figure 10 shows the fracture surfaces and longitudinal microstructures of the stress rupture samples tested at 975 °C/225 MPa. As seen from the Figure 10a–d, there was no obvious dendrite morphology on the fracture surface of the alloy, which indicated that the crack was not formed and propagated at the interdendritic region during the stress rupture. The SEM image of the longitudinal microstructures of the alloy showed that the crack initiated and propagated at the grain boundary and thus caused intergranular failure of the superalloys. Therefore, the stress rupture properties were attributed to the strength and stability of the grain boundary. Microcracks in Alloy I exhibited the tendency to connect with each other to form long intergranular cracks, which would deteriorate the stress rupture properties. In contrast, the microcracks in Alloy II and Alloy III were hard to get connected to form the long cracks, as shown in Figure 10f,g. The number of the carbides near the grain boundary increased with the addition of Mo, which would restrain the extension of microcracks and improve the stress rupture life of the high-W superalloys. Furthermore, the increase in Mo content led to a larger negative misfit of the γ and γ′ phases in the high-W superalloys. The previous research reported that a large magnitude of misfit would induce dense γ/γ′ interface dislocation networks and reduce the minimum creep rates during the creep tests at high temperature [39,40]. In addition, the γ and γ′ microstructures became rafted as the creep deformation continued, and no TCP phase was found in the microstructures in the high-W superalloys with low Mo content, which indicated that the microstructures of the superalloys were relatively stable during high-temperature deformation, while a large number of needle-like TCP phases were precipitated in Alloy IV after the heat treatment, which would promote the initiation and propagation of the crack due to the stress concentration at the tip of the TCP phases during the deformation process, as shown in Figure 10h; similar results were also reported by Tian [41]. Meanwhile, a large number of TCP phases precipitated in the microstructures, consuming the solid solution strengthening elements W and Mo, which resulted in the decrease in strength [42].

## 4. Conclusions

In summary, the effects of Mo addition on the microstructures and mechanical properties of high-W superalloys were investigated in this work. The following conclusions can be obtained:

(1) A needle-like TCP phase was formed in the alloy with high Mo content in contrast to the other three alloys with low Mo content. The Mo addition increased the lattice parameter of the γ and γ′ phases and also changed the lattice misfit of the γ and γ′ phases towards a larger negative.

(2) The mechanical property analysis indicated that the addition of Mo improved the yield strength at room temperature due to the solid solution strengthening and coherency strengthening. The precipitated TCP consumed the solid solution strengthening elements W and Mo, which deteriorated the ultimate tensile strength.

(3) The improvement of the stress rupture lives at 975 °C/225 MPa was the result of the combination of the carbides suppressing the link-up and extension of the microcracks and a more negative misfit with the increase in Mo content. The TCP phases precipitated and deteriorated the strength of the interdendritic region of Alloy IV, which resulted in the decrease in the stress rupture lives.

## Figures and Tables

**Figure 1 materials-15-07509-f001:**
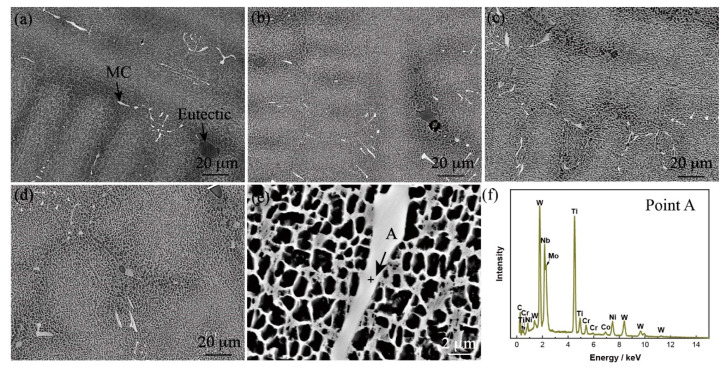
Microstructures of the as-cast high-W superalloys with different Mo content: (**a**) Alloy I, (**b**) Alloy II, (**c**) Alloy III, (**d**) Alloy IV, (**e**) magnified image of Alloy IV, (**f**) EDS result corresponds to the point A in (**e**).

**Figure 2 materials-15-07509-f002:**
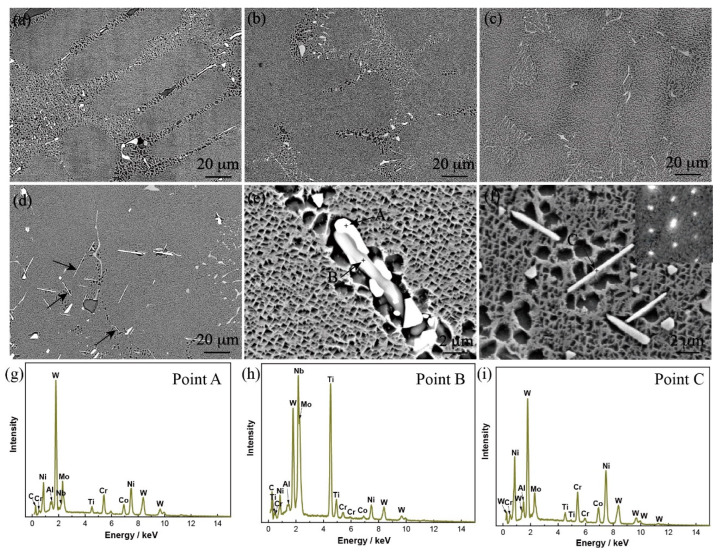
Microstructures of the heat-treated alloys with different Mo content: (**a**) Alloy I, (**b**) Alloy II, (**c**) Alloy III, (**d**) Alloy IV, (**e**,**f**) magnified images of Alloy IV alloy; (**g,h,i**) EDS results correspond to the points A, B, and C in (**e**,**f**). (The inset in (**f**) corresponds to the selected area electron diffraction pattern).

**Figure 3 materials-15-07509-f003:**
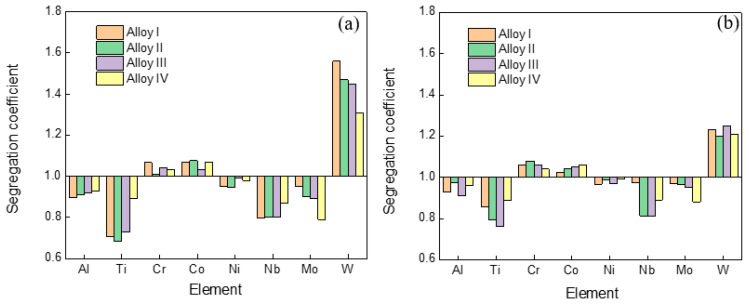
Segregation coefficient *k* of different elements in the alloys with different Mo contents: (**a**) cast state, (**b**) heat-treated state.

**Figure 4 materials-15-07509-f004:**
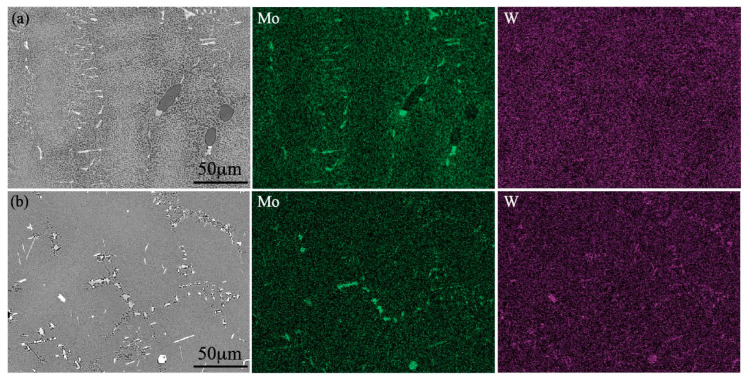
Distribution of Mo and W in Alloy IV: (**a**) cast state, (**b**) heat-treated state.

**Figure 5 materials-15-07509-f005:**
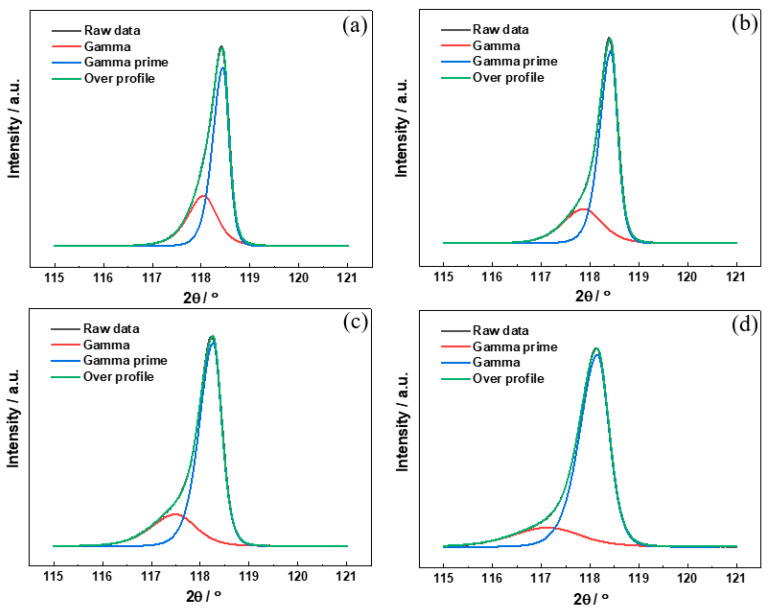
Profiles of (004) diffraction peaks and peak fitting results of the single-crystal alloys with different Mo contents: (**a**) Alloy I, (**b**) Alloy II, (**c**) Alloy III, (**d**) Alloy IV.

**Figure 6 materials-15-07509-f006:**
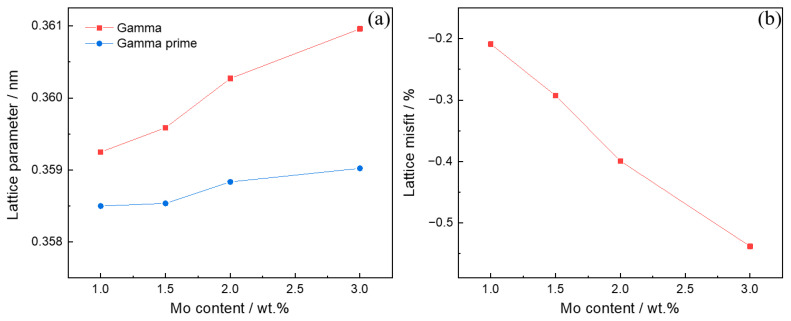
(**a**) Lattice parameters of the γ and γ′ phases and (**b**) lattice misfit of the alloys with different Mo contents.

**Figure 7 materials-15-07509-f007:**
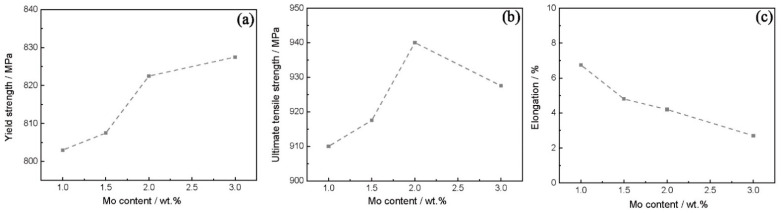
The tensile properties of the alloys with different Mo contents at room temperature: (**a**) yield strength, (**b**) ultimate tensile strength, (**c**) elongation.

**Figure 8 materials-15-07509-f008:**
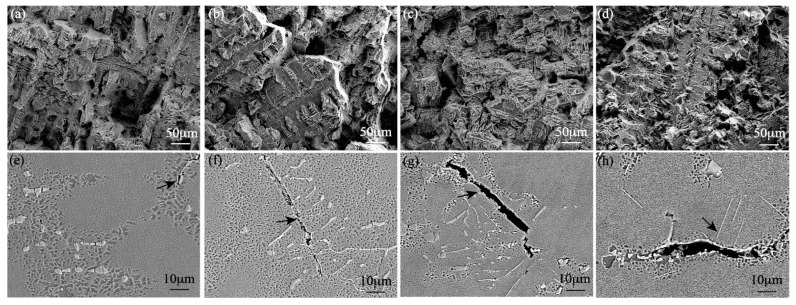
SEM images of tensile fracture surfaces and the longitudinal microstructure at room temperature: (**a**,**e**) Alloy I, (**b**,**f**) Alloy II, (**c**,**g**) Alloy III, (**d**,**h**) Alloy IV.

**Figure 9 materials-15-07509-f009:**
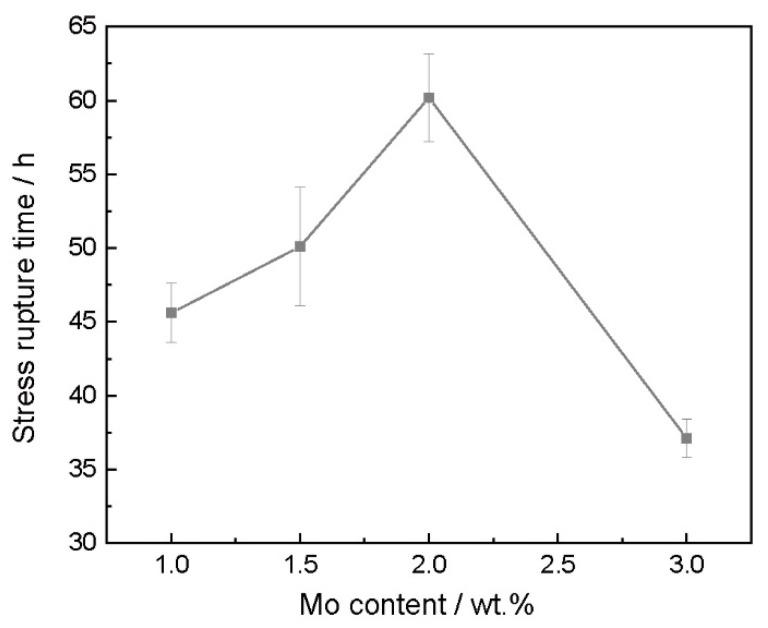
The stress rupture life of the alloys with different Mo content at 975 °C/225 MPa.

**Figure 10 materials-15-07509-f010:**
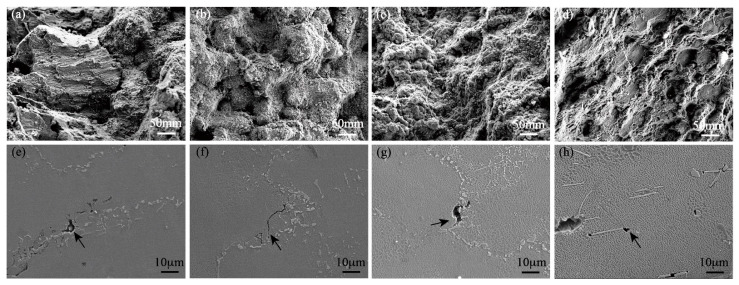
SEM images of fracture surfaces and the longitudinal microstructures after stress rupture testing at 975 °C/225 MPa: (**a**,**e**) Alloy I, (**b**,**f**) Alloy II, (**c**,**g**) Alloy III, (**d**,**h**) Alloy IV.

**Table 1 materials-15-07509-t001:** Chemical compositions of experimental alloys (mass fraction, wt.%).

NO.	W	Mo	Al	Co	Cr	Nb	Ti	Zr	B	C	Ni
Alloy I	10	1.0	5.5	10.0	9.3	1	2.5	0.03	0.015	0.18	Bal.
Alloy II	10	1.5	5.5	10.0	9.3	1	2.5	0.03	0.015	0.18	Bal.
Alloy III	10	2.0	5.5	10.0	9.3	1	2.5	0.03	0.015	0.18	Bal.
Alloy IV	10	3.0	5.5	10.0	9.3	1	2.5	0.03	0.015	0.18	Bal.

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
