# Peer review of "Effect of Mo on the Microstructures and Mechanical Properties of the Polycrystalline Superalloy with High W Content"

_materials, 2022, doi:10.3390/ma15217509_

Round 1
Reviewer 1 Report
The current manuscript is a study on the microstructure and mechanical properties of a polycrystalline Ni-based superalloy with high W content. The effect of Mo addition from 1 to 3 wt.% was systematically studied. TCP phases were found to appear in the alloy with 3 wt.% Mo addition, and the associated changes in the microstructure, lattice parameter misfit, and mechanical properties were discussed. The paper is basically well-written. The present results can provide good reference for researchers in the same field. It can be recommended for publication after some minor language polishing. Minor comment: please refine the grammar on lines 259, 263, and 265.Author Response
Response:
The language in the manuscript was carefully polished.
In contrast, the microcracks in Alloy II and Alloy III were hard to get connected to form the long cracks, as shown in Figure 10f-g.
Furthermore, the increase of Mo content lead to a larger negative misfit of γ and γ′ phase in the high W superalloys.
The previous researches reported that a large magnitude of misfit would induce dense γ/γ′ interfaces dislocation networks and reduce the minimum creep rates during the creep tests at high temperature

Reviewer 2 Report
The manuscript can be accepted after addressing the following comments:
1) Please add more recently published literature, specifically related to Mo based super alloys if possible.
2) What about the dimensions of the tensile sample?
3) Which ASTM standard was followed for the test?
4) Lines 109-111 needs references.
5) What huge peak of Ti on Fig.1f reflects?
6) Why no data is shown for alloy I and II in Fig.3?
7) Fig. 5 is not clear?
Author Response
The manuscript can be accepted after addressing the following comments:
1) Please add more recently published literature, specifically related to Mo based super alloys if possible.
Response:
The recently references were cited in the revised manuscript.
- Wei, B.; Lin, Y.; Huang, Z.; Huang, L.; Zhou, K.; Zhang, L.; Zhang, L., A novel Re-free Ni-based single-crystal superalloy with enhanced creep resistance and microstructure stability. Acta Mater. 2022, 240, 118336.
- Sun, S.; Sheng, N.; Fan, S.; Ma, Y.; Cao, X.; Sang, Z.; Hou, G.; Li, J.; Zhou, Y.; Sun, X., Abnormal feather-like grains induced by the thin-wall effect in a polycrystalline nickel-based superalloy. J. Alloy. Compd. 2022, 901, 163581.
- Guo, X.; Antonov, S.; Lu, F.; Zheng, W.; Yuan, X.; Cormier, J.; Feng, Q., Solidification rate driven microstructural stability and its effect on the creep property of a polycrystalline nickel-based superalloy K465. Mater. Sci. Eng. A 2020, 770, 138530.
- Liu, S.; Wang, C.; Yan, P.; Yu, T., The effect of Ta, W, and Re additions on the tensile-deformation behavior of model Ni-based single-crystal superalloys at intermediate temperature. Mater. Sci. Eng. A 2022, 850, 143594.
- Gong, L.; Chen, B.; Du, Z.; Zhang, M.; Liu, R.; Liu, K., Investigation of solidification and segregation characteristics of cast Ni-base superalloy K417G. J. Mater. Sci. Technol. 2018, 34, (3), 541-550.
- Sun, J.; Liu, J.; Li, J.; Chen, C.; Wang, X.; Zhou, Y.; Sun, X., Dual effects of Ru on the microstructural stability of a single crystal superalloy. Scripta Mater. 2021, 205, 114209.
2) What about the dimensions of the tensile sample?
Response:
The dimensions of the tensile sample were provided in the revised manuscript.
Specimens with a diameter of 5 mm and a gage length of 30 mm were machined for the tensile tests, and specimens with a diameter of 5 mm and a gauge length of 25 mm were machined for the stress rupture tests.
3) Which ASTM standard was followed for the test?
Response:
The test method and the dimensions of the samples were followed the Chinese standard HB 5143-96.
4) Lines 109-111 needs references.
Response:
The relative reference was cited in the revised manuscript.
Generally, the content of eutectic in the alloy is related to the degree of segregation of elements during the solidification process [25].
- Gong, L.; Chen, B.; Du, Z.; Zhang, M.; Liu, R.; Liu, K., Investigation of solidification and segregation characteristics of cast Ni-base superalloy K417G. J. Mater. Sci. Technol. 2018, 34, (3), 541-550.
5) What huge peak of Ti on Fig.1f reflects?
Response:
The MC carbides are consist of C and other metallic elements, such as Mo, Ti, Nb, W and so on. Thus, the huge peak of Ti on Fig.1f reflects was corresponding to the Ti in MC carbides.
6) Why no data is shown for alloy I and II in Fig.3?
Response:
The data for Alloy I and Alloy II were provided in the revised manuscript.
7) Fig. 5 is not clear?
Response:
The clear images were provided in the revised manuscript.

Reviewer 3 Report
Title: ‐ Effect of Mo on the Microstructures and Mechanical Properties of the Polycrystalline Superalloy with High W Content (ID: materials-1862832).
In this manuscript, the authors studied the effect of Mo on the polycrystalline superalloy using a new method, vacuum induction melting (VIM). The microstructure characterizations and tensile properties were further investigated. The experimental was clearly described, and the conclusions were solid supported by the experimental results. Therefore, I can suggest its publication after considering following minor comments. What was the selection parameter for solution heat treatment?
- Application wise, the aim of the research is to seek the potential to use these high W content Ni-based superalloys is widely used in the production of engine turbine blades. Besides comparison on the properties between the studied Ni-based alloys and other alloys, it would be good to provide any advantage of this Ni-based superalloys cost-vise.
2. What is the motivation?
3. Digital camera image of experimental setup need to be included in the experimental section.
4. Please mention what is the standard shape and size of the SEM and tensile test specimen.
5. What is the purpose of solution treatment in the present work?
6. Missing the scale bar in Figure 1 (b,c and d), Figure 2 (a, b and c).
7. Tensile engineering stress-strain curves of the developed Mo doped polycrystalline superalloys and failed specimen need to be incorporated.
8. Do you have any comparable study on the same composite by other sintering methods? If yes, please compare with them.
. There are few grammatical mistakes Authors are requested to proof read complete manuscript and check for English grammar.
Author Response
- Application wise, the aim of the research is to seek the potential to use these high W content Ni-based superalloys is widely used in the production of engine turbine blades. Besides comparison on the properties between the studied Ni-based alloys and other alloys, it would be good to provide any advantage of this Ni-based superalloys cost-vise.
Response:
Thanks for the valuable comments. We would conduct efforts to extend the application of high W content Ni-based superalloys due to the low cost and high performance of the series superalloys.
- What is the motivation?
Response:
As mentioned in the Introduction, the high W superalloys have the advantage of the low cost and high performance, while the macrosegregation is prone to appear in the high W superalloys due to the large solidification segregation coefficient of W element, especially in the superalloys with 14wt.% W and upwards[14, 15]. As compared with W, Mo had the similar strengthening effect, substituting Mo for W would contribute to the decrease of macrosegregation and the density of the superalloys. However, there is few further detailed and systemic researches on the effect of Mo on the polycrystalline superalloy with high W content. Hence, the effect of Mo was studied to provide more information for developing high strength high W superalloys.
- Digital camera image of experimental setup need to be included in the experimental section.
Response:
The experimental setups in this manuscript are conventional equipment for preparing superalloy specimens, characterizing microstructures and performing the mechanical tests [1].
- Gong, L.; Chen, B.; Zhang, L.; Ma, Y.; Liu, K., Effect of cooling rate on microstructure, microsegregation and mechanical properties of cast Ni-based superalloy K417G. J. Mater. Sci. Technol. 2018, 34, (5), 811-820.
- Please mention what is the standard shape and size of the SEM and tensile test specimen.
Response:
The dimensions of the tensile sample and SEM specimens were provided in the revised manuscript.
Specimens with a diameter of 5 mm and a gage length of 30 mm were machined for the tensile tests, and specimens with a diameter of 5 mm and a gauge length of 25 mm were machined for the stress rupture tests.
Specimens with a diameter of 5 mm and a height of 3 mm were prepared for SEM observations.
Longitudinal section samples with a length of 10 mm, a width of 4 mm and a thickness of 1 mm were cut from the fractured specimens.
- What is the purpose of solution treatment in the present work?
Response:
The purpose of the solution treatment is to reduce the segregation behavior between the dendritic and interdendritic regions, which was usually performed to improve the mechanical properties [2].
- Hegde, S. R.; Kearsey, R. M.; Beddoes, J. C., Designing homogenization–solution heat treatments for single crystal superalloys. Mater. Sci. Eng. A 2010, 527, (21), 5528-5538.
- Missing the scale bar in Figure 1 (b,c and d), Figure 2 (a, b and c).
Response:
The scale bars were provided in the revised manuscript.
- Tensile engineering stress-strain curves of the developed Mo doped polycrystalline superalloys and failed specimen need to be incorporated.
Response:
The engineering stress-strain curves were not accurate because of not using the extensometer during the tensile tests. Thus, the curves were not included in the manuscript. The ductility (δ) was calculated by measuring the gage length (lf) after fracture.
(1)
where l0 is the initial gage length (30 mm).
- Do you have any comparable study on the same composite by other sintering methods? If yes, please compare with them.
Response:
We have not performed the other sintering methods up till the present moment, and we will attempt to conduct the other method, such as powder metallurgy.
- There are few grammatical mistakes Authors are requested to proof read complete manuscript and check for English grammar.
Response:
The language in the manuscript was carefully polished

Round 2
Reviewer 3 Report
Dear Sir,
Please accept in present form.
Regards
Author Response
We thank the reviewer for the valuable comments. The language in the manuscript was carefully polished again.